# Enhanced Anti-Inflammatory and Non-Alcoholic Fatty Liver Disease (NAFLD) Improvement Effects of *Bacillus subtilis*-Fermented *Fagopyrum tataricum* Gaertner

**Chan-Hwi Park** **, Hyun Kang * and Sung-Gyu Lee ***

Department of Medical Laboratory Science, College of Health Science, Dankook University, Cheonan-si 31116, Republic of Korea; cksgnl1014@naver.com
* Correspondence: hkang@dankook.ac.kr (H.K.); sung-gyu@dankook.ac.kr (S.-G.L.)

**Abstract:** In this study, we investigated the enhanced anti-inflammatory activity and the effects on non-alcoholic fatty liver disease (NAFLD) of fermented *Fagopyrum tataricum* (*F. tataricum*) Gaertner extract (FFT) through in vitro analysis. We utilized high-performance liquid chromatography (HPLC) to analyze the non-fermented *F. tataricum* Gaertner extract (NFT) and the marker components, rutin and quercetin in FFT, to confirm changes in composition due to fermentation. The anti-inflammatory activity of NFT and FFT was evaluated using a lipopolysaccharide (LPS)-induced RAW 264.7 cell inflammation model. Simultaneously, the NAFLD improvement effects were measured by evaluating lipid accumulation and the expression of lipid synthesis regulators in free fatty acid (FFA)-induced HepG2 cells. HPLC analysis confirmed an increase in rutin content after the fermentation of *F. tataricum* Gaertner. Upon treatment with NFT and FFT at a concentration of 400 μg/mL, LPS-induced nitric oxide (NO) production values in RAW 264.7 cells were reduced to 16.12 μM and 2.09 μM, respectively, indicating enhanced significant inhibition ($p < 0.05$) of NO production through fermentation. FFT demonstrated the significant inhibition ($p < 0.05$) of inducible nitric oxide synthase (iNOS), cyclooxygenase-2 (COX-2) protein, and inflammatory cytokine mRNA expression through the nuclear factor kappa B (NF-κB) and mitogen-activated protein kinase (MAPK) pathways in LPS-induced RAW 264.7 cells. In FFA-induced HepG2 cells, FFT significant suppressed ($p < 0.05$) lipid accumulation and the expression of sterol regulatory element binding protein (SREBP)-1c, CCAAT/enhancer binding protein (C/EBP)α proteins, and acetyl-CoA carboxylase (ACC) mRNA. The results of this study suggest the potential utilization of FFT as a material for improving NAFLD.

**Keywords:** *Fagopyrum tataricum* Gaertner; non-alcoholic fatty liver disease; anti-inflammatory; bioconversion; *Bacillus subtilis*



## 1. Introduction

Non-alcoholic fatty liver disease (NAFLD) encompasses a spectrum of conditions ranging from simple steatosis, characterized by the excessive accumulation of fat in liver cells, to non-alcoholic steatohepatitis (NASH), which involves liver cell necrosis, inflammation, and fibrosis, and may progress to advanced stages such as cirrhosis [1–4]. The prevalence of NAFLD is on the rise not only in several Western countries but also in East Asia, reflecting an increase in the obese population. Globally, it is reported to affect approximately 20–30% of the world's population, varying among nations [5–7].

Inflammation, an immune response arising from infections, toxins, and other factors, can lead to inflammatory diseases when excessive production of inflammatory mediators occurs. When reactive oxygen species (ROS) are excessively generated as by-products of cellular metabolic activities in the body, they can initiate the inflammatory response, inducing the production of inflammatory cytokines [8–13].

Currently, treatments for NAFLD primarily focus on weight loss and the use of blood sugar-lowering agents. However, there is a growing need for targeted therapies specifically designed for NAFLD as the prevalence of this condition continues to rise [14].

*Fagopyrum esculentum* (*F. esculentum*), belonging to the genus *Fagopyrum* and the family Polygonaceae, is cultivated globally and has garnered attention as a health food due to its high content of polysaccharides, essential amino acids, phenolic acids, unsaturated fatty acids, and flavonoids [15]. Two main varieties of *F. esculentum* are cultivated: *F. esculentum* and *Fagopyrum tataricum* (*F. tataricum*) Gaertner. *F. esculentum* is predominantly cultivated in Asian regions such as South Korea and Japan, while *F. tataricum* Gaertner is grown in high-altitude areas like the Himalayas and Nepal [16]. *F. tataricum* Gaertner is rich in flavonoid components such as rutin, quercetin, quercetrin, and catechins, exhibiting various beneficial effects such as antioxidant, anti-inflammatory, antidiabetic, lipid suppression, and skin-whitening properties. Among these, rutin is reported to have functional attributes including antioxidant, anti-inflammatory, anticancer, and antidiabetic activities [17–21]. In addition to the bioactive components such as phenolic compounds in buckwheat, bioactive carbohydrates/polysaccharides and peptides play crucial roles in health promotion. Polysaccharides, found abundantly in natural sources like plants, exhibit various bioactivities such as immunomodulation, antioxidant, and anti-inflammatory effects [22,23]. They have garnered attention in functional food research due to their potential in preventing and managing chronic diseases, including liver conditions like NAFLD [24].

Rutin, a key component of *F. tataricum* Gaertner, is a substance wherein rutinoside is attached to quercetin [17,18]. It has been reported to have a higher rutin content compared to *F. tataricum*. Rutin is currently being researched in various fields due to its functional properties. Although various studies have been conducted on *F. tataricum* Gaertner, there has still been limited research on its application in bioconversion technology [25]. Therefore, in this study, we investigated the mechanism of NAFLD improvement in vitro using fermented *F. tataricum* Gaertner, a material utilizing bioconversion technology.

## 2. Materials and Methods

### 2.1. Materials

The *F. tataricum* Gaertner used in this study was purchased from Pyeongchang, Gangwon-do (Korea). *Bacillus subtilis* (KCTC 3014), employed for fermentation, was obtained from the Korea Research Institute of Bioscience and Biotechnology (KRIBB) Biological Resource Center (KCTC, Daejeon, South Korea). Filter paper (Whatman No. 3, Whatman Ltd., Buckinghamshire, UK) was purchased from Whatman. Rutin, quercetin, oleic acid, palmitic acid, oil red O powder, Griess reagent, lipopolysaccharide (LPS), formaldehyde, and nitric oxide (NO) were procured from Sigma-Aldrich Co. (St. Louis, MO, USA). Thiazolyl blue tetrazolium bromide (MTT) and phosphate-buffered saline (PBS) were obtained from Biosaesang (Yongin-si, Republic of Korea). Phosphoric acid was sourced from Junsei (Tokyo, Japan), acetonitrile from J.T Baker (Phillipsburg, NJ, USA), TRIzol reagent, and DEPC-treated water from Ambion (Austin, TX, USA). Dimethyl sulfoxide (DMSO) was from Daejung (Busan, Republic of Korea), isopropanol from Duksan (Ansan-si, Republic of Korea), and methanol, chloroform, and ethanol from Samchun Chemical Co. (Seoul, Republic of Korea). Nuclear and cytoplasmic extraction reagent (NE-PER) and ECL kit were purchased from Thermo Fisher Scientific (Waltham, MA, USA). The chemiluminescence imaging system used was from ATTO (Motoasakusa, Taito-ku, Tokyo, Japan). Nano Drop equipment was obtained from Thermo Fisher Scientific (Waltham, MA, USA) and PCR was conducted using Bio-Rad equipment (CFX96, Hercules, CA, USA). The microplate spectrophotometer utilized was from Bio-Rad (xMARK, Hercules, CA, USA).

### 2.2. Preparation of NFT and FFT

A quantity of 100 g of dried *F. tataricum* Gaertner grain was ground and used in the experiment. Subsequently, 10 times the amount of distilled water (DW) was added for rehydration. The mixture underwent sterilization using an autoclave. Although sterilization during fermentation can potentially degrade compounds, it is necessary to remove

contaminants other than fermentation strains. *Bacillus subtilis* (KCTC 3014) was cultured in a broth medium (Kisanbio Co., Seoul, Republic of Korea) with a pH of 7.2. These microorganisms were cultured at 30 °C and used for the experiment. They were inoculated at a 10% concentration. The fermentation was conducted at 37 °C for 3 days. After fermentation, the extract was obtained by using 70% ethanol for 24 h. The extracts were obtained after removing the bacteria through the filtration process, concentrated under reduced pressure at 55 °C, and finally freeze-dried for use in the experiment. The non-fermented *F. tataricum* Gaertner (NFT), used as a control, was prepared by inoculating the medium and allowing it to stand under the same conditions for 3 days before extraction. The process of preparing the extracts is illustrated in Figure 1.

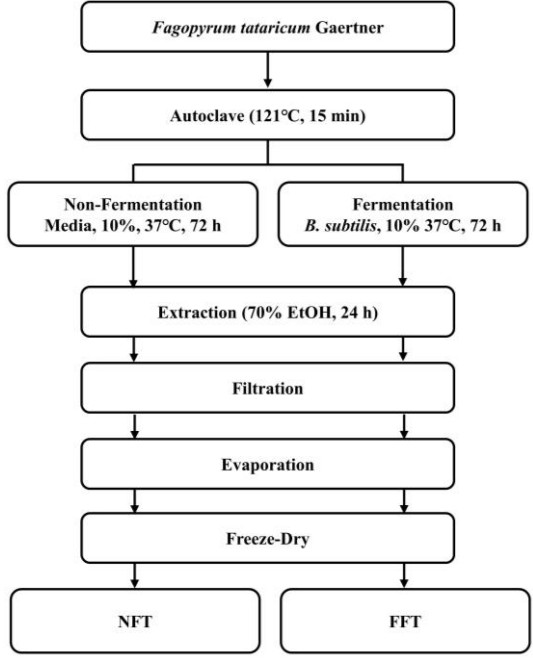

**Figure 1.** Diagram for preparation of NFT and FFT from *F. tataricum* Gaertner.

### 2.3. High-Performance Liquid Chromatography (HPLC)

HPLC was employed to analyze the components rutin (Sigma-Aldrich Co.) and quercetin (Sigma-Aldrich Co.) in NFT and FFT. The analysis employed the Waters 2690 HPLC system (Waters Corporation, Milford, MA, USA). Samples were dissolved in DMSO, and a ZORBAX SB-C$_{18}$ column (4.6 × 250 mm, 5 μm) was used. The column temperature was maintained at room temperature (RT), and a Diode Array Detector (PDA) was employed. For quercetin analysis, the mobile phase A consisted of 0.1% phosphoric acid dissolved in DW and mobile phase B was acetonitrile. For rutin analysis, mobile phase A was methanol and mobile phase B was 0.5% phosphoric acid dissolved in DW. The HPLC analysis conditions are presented in Table 1.

**Table 1.** Conditions for the HPLC analysis of the quercetin and rutin content in the NFT and FFT.

| Instrument | | Conditions | | |
|---|---|---|---|---|
| Flow rate | | 1.0 mL/min | | |
| Injection volume | | 10 μL | | |
| Detector | | Diode Array Detector (254 nm) | | |
| | Time (min) | Quercetin | 0.1% Phosphoric acid in DW | Acetonitrile |
| | | Rutin | Methanol | 0.5% Phosphoric acid in DW |
| Mobile phase (Gradient) | 0 | | 30 | 70 |
| | 5 | | 35 | 65 |
| | 10 | | 40 | 60 |
| | 15 | | 50 | 50 |
| | 18 | | 30 | 70 |
| | 25 | | Stop | |

### 2.4. Cell Lines and Cultivation

RAW 264.7 cells, derived from mice, and HepG2 cells, derived from human hepatocellular carcinoma, were obtained from the Korean Cell Line Bank (KCLB, Seoul, Republic of Korea). The cells were cultured using DMEM medium supplemented with 10% FBS and 1% penicillin-streptomycin. Cultivation was performed at 37 °C in a 5% $CO_2$ environment, with subculturing conducted at 2–3 day intervals.

### 2.5. Cell Viability Assessment

Cell viability was measured through MTT assay [26]. RAW 264.7 cells were seeded in a 96-well plate at a concentration of $5 \times 10^4$ cells/well and cultured for 24 h. After removing the medium, cells were treated with NFT and FFT prepared at different concentrations in serum-free medium containing LPS (100 ng/mL). After 24 h of incubation, MTT solution (5.0 mg/mL) was added, and the cells were further incubated for 3 h under the same culture conditions. The resulting formazan crystals were dissolved through the addition of 100 μL of DMSO to each well, followed by 10 min of shaking. The absorbance was measured at the wavelength of 550 nm. HepG2 cells were seeded in a 24-well plate at a concentration of $1 \times 10^5$ cells/well and cultured for 24 h. To create a cellular model of fatty acid-induced NAFLD, cells were treated with various concentrations of FFT prepared in serum-free medium containing free fatty acid (FFA, oleic acid (2): palmitic acid (1), 1 mM) for 24 h [27]. Cell viability was measured through MTT assay.

### 2.6. Nitric Oxide (NO) Production Assay

NO production was measured according to the method of Green et al. [28]. RAW 264.7 cells were cultured in a 96-well plate at a concentration of $5 \times 10^4$ cells/well for 24 h. After removing the medium, cells were treated with NFT and FFT prepared at different concentrations in serum-free medium containing LPS (100 ng/mL). After 24 h, Griess reagent was added to the culture supernatant for 10 min, and absorbance was measured at 540 nm.

### 2.7. Reverse Transcription–Polymerase Chain Reaction (RT-PCR)

RT-PCR was performed according to the method of Rio [29]. Total RNA from RAW 264.7 cells and HepG2 cells was extracted using Trizol. The extracted RNA was used to synthesize cDNA under the conditions of 42 °C for 1 h and 94 °C for 5 min using Bioneer's RT premix. The synthesized DNA was then used for PCR using Bioneer's PCR premix with the addition of 5 pmole of primers. PCR products were visualized on a 2% gel. The primer sequences used for PCR are provided in Table 2.

**Table 2.** The sequences of primers used for PCR.

| Gene | Origin | | Sequence |
|------|--------|---------|----------|
| iNOS | mouse | Forward | 5′-CTTGCAAGTCCAAGTCTTGC-3′ |
| | | Reverse | 5′-GTATGTGTCTGCAGATGTGCTG-3′ |
| COX-2 | mouse | Forward | 5′-ACATCCCTGAGAACCTGCAGT-3′ |
| | | Reverse | 5′-CCAGGAGGATGGAGTTGTTGT-3′ |
| IL-1β | mouse | Forward | 5′-CATATGAGCTGAAAGCTCTCCA-3′ |
| | | Reverse | 5′-GACACAGATTCCATGGTGAAGTC-3′ |
| IL-6 | mouse | Forward | 5′-GGAGGCTTAAITACACATGTT-3′ |
| | | Reverse | 5′-TGATTCAAGATGAATTGGAT-3′ |
| TNF-α | mouse | Forward | 5′-TTCGAGTGACAAGCCTGTAGC-3′ |
| | | Reverse | 5′-AGATTGACCTCAGCGCTGAGT-3′ |
| GAPDH | mouse | Forward | 5′-CCAGTATGACTCCACTCACG-3′ |
| | | Reverse | 5′-CCTTCCACAATGCCAAGTT-3′ |
| SREBP-1 c | Human | Forward | 5′-CAGTGGAGGGAACACAGACG- 3′ |
| | | Reverse | 5′-AAAGACTGGGCTGTCAGGCT- 3′ |
| PPARγ | Human | Forward | 5′-CAGGAGCAGAGCAAAGAGGTG-3′ |
| | | Reverse | 5′-CAAACTCAAACTTGGGCTCCA-3′ |
| ACC1 | Human | Forward | 5′-GGAACAGTGTGCGGTGAAAC- 3′ |
| | | Reverse | 5′-TCACTAGTGATCCGAGCAGC-3′ |
| GAPDH | Human | Forward | 5′-CGGAGTCAACGGATTTGGTCGTAT-3′ |
| | | Reverse | 5′-AGCCTTCTCCATGGTGGTGAAGAC-3′ |

### 2.8. Western Blot

Western blot assay was performed according to the method of Kim [30]. RAW 264.7 cells were seeded in a 6-well plate at a concentration of $5 \times 10^5$ cells/well and incubated for 24 h. After removing the medium, cells were treated with FFT prepared at different concentrations in serum-free medium containing LPS (100 ng/mL). After 24 h, cells were lysed, and proteins were obtained from the supernatant. HepG2 cells were treated similarly with FFT in FFA-containing medium. For mitogen-activated protein kinase (MAPK) pathway analysis, samples were pretreated for 24 h, followed by a 30 min treatment with LPS. Nuclear and cytoplasmic fractions were obtained using NE-PER. Proteins were separated on a 10% SDS-PAGE, transferred to a PVDF membrane, blocked with 5% skim milk, and then incubated with primary and secondary antibodies. Protein bands were visualized using an ECL kit and a chemiluminescence imaging system.

### 2.9. Oil Red O Stain

Oil red O staining was performed according to the method of Yao et al. [27]. HepG2 cells were seeded in a 6-well plate at a concentration of $5 \times 10^5$ cells/well and cultured for 24 h. After removing the medium, cells were treated with FFT prepared at different concentrations in serum-free medium containing FFA (1 mM). After 48 h, cells were fixed, stained with oil red O, and observed under a microscope. Intracellular lipid content was measured at 520 nm after solubilization in isopropanol.

### 2.10. Statistical Analysis

All experimental results are presented as means ± standard deviations. Statistical analysis was performed using ANOVA, and significance was assessed by Duncan's multiple range test with a significance level of $p < 0.05$.

## 3. Results

### 3.1. Rutin and Quercetin Content in NFT and FFT

HPLC analysis was conducted to measure the changes in rutin and quercetin content in NFT and FFT during the bioconversion process. The quantification assays were conducted three times. Statistical analysis was applied to identify significant differences between the samples. Quercetin peaks were detected around 23 min while rutin peaks were observed around 2 min. The calculated quercetin content revealed that NFT contained 10.69 ± 0.02% quercetin while FFT contained 5.32 ± 0.01% quercetin (Figure 2A). After fermentation, there was a significant reduction in quercetin content. Rutin content analysis showed that NFT had 5.34 ± 0.01% rutin and FFT had 7.10 ± 0.02% rutin (Figure 2B). Interestingly, rutin content increased in FFT compared to NFT.

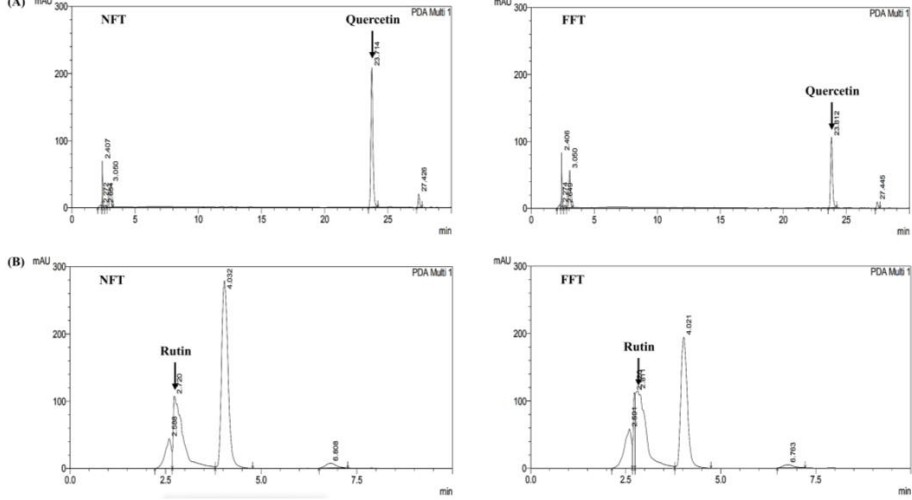

**Figure 2.** Contents of (**A**) quercetin and (**B**) rutin in NFT and FFT.

### 3.2. Anti-Inflammatory Activity by NFT and FFT in LPS-Induced RAW 264.7 Cells

To assess the cytotoxicity of NFT and FFT, RAW 264.7 cells were treated with various concentrations (25, 50, 100, 200, and 400 μg/mL), and cell viability was measured. The results indicated no significant toxicity (Figure 3A). NO production, measured using the Griess assay, revealed a concentration-dependent decrease in NO levels in both NFT and FFT. FFT showed a substantial reduction in NO production compared to NFT (Figure 3B).

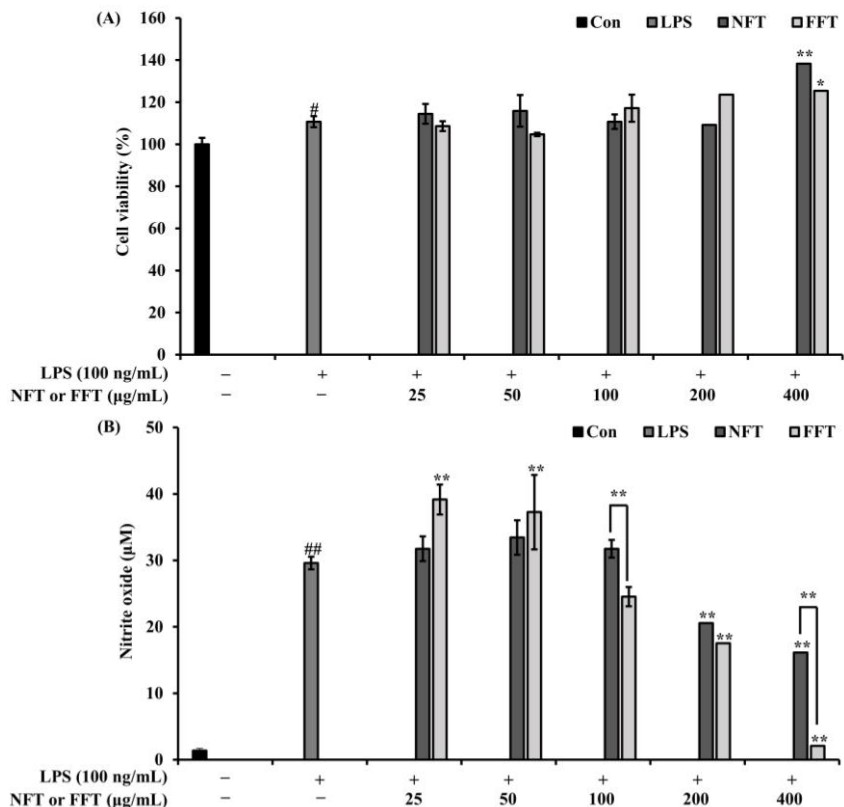

**Figure 3.** Effects of NFT and FFT on cell viability and NO production in LPS-induced RAW 264.7 cells. RAW 264.7 cells were treated with different concentrations (25, 50, 100, 200, and 400 μg/mL) of each sample and LPS (100 ng/mL) for 24 h. (**A**) Cell viability was measured through MTT assay. (**B**) NO contents in cell supernatant were measured through Griess reagent assay. Data are expressed as means ± standard deviations ($n = 3$). # $p < 0.05$ and ## $p < 0.01$ compared to the normal group. * $p < 0.05$ and ** $p < 0.01$ compared to the LPS group.

### 3.3. Effect of FFT on iNOS and COX-2 Protein Expression

Western blot analysis was conducted to examine the expression of inducible nitric oxide synthase (iNOS) and cyclooxygenase-2 (COX-2) in LPS-induced RAW 264.7 cells treated with different concentrations (25, 50, 100, 200, and 400 μg/mL) of FFT. The results showed a concentration-dependent decrease in iNOS and COX-2 expression in FFT-treated groups compared to the LPS-treated group (Figure 4).

### 3.4. Effect of FFT on Inflammatory Cytokine Expression

RNA was extracted from LPS-stimulated RAW 264.7 cells treated with various concentrations (25, 50, 100, 200, and 400 μg/mL) of FFT, and the expression of inflammatory cytokines was assessed. FFT treatment resulted in a concentration-dependent reduction in the expression of inflammatory cytokines compared to the LPS-treated group (Figure 5).

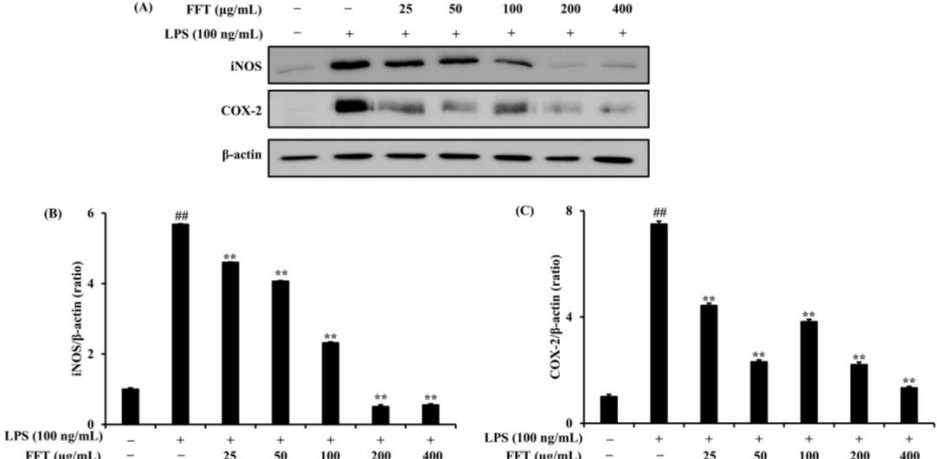

**Figure 4.** Effects of FFT on iNOS and COX-2 protein expression in LPS-induced RAW 264.7 cells. RAW 264.7 cells were treated with various concentrations (25, 50, 100, 200, and 400 μg/mL) of FFT and LPS (100 ng/mL). The results indicate (**A**) iNOS and COX-2 protein expression levels in FFT. The graph indicates ratios of (**B**) iNOS and (**C**) COX-2. β-actin was used as a loading control. Data are expressed as means ± standard deviations (*n* = 3). ## *p* < 0.01 compared to the normal group. ** *p* < 0.01 compared to the LPS group. See also Figure S1.

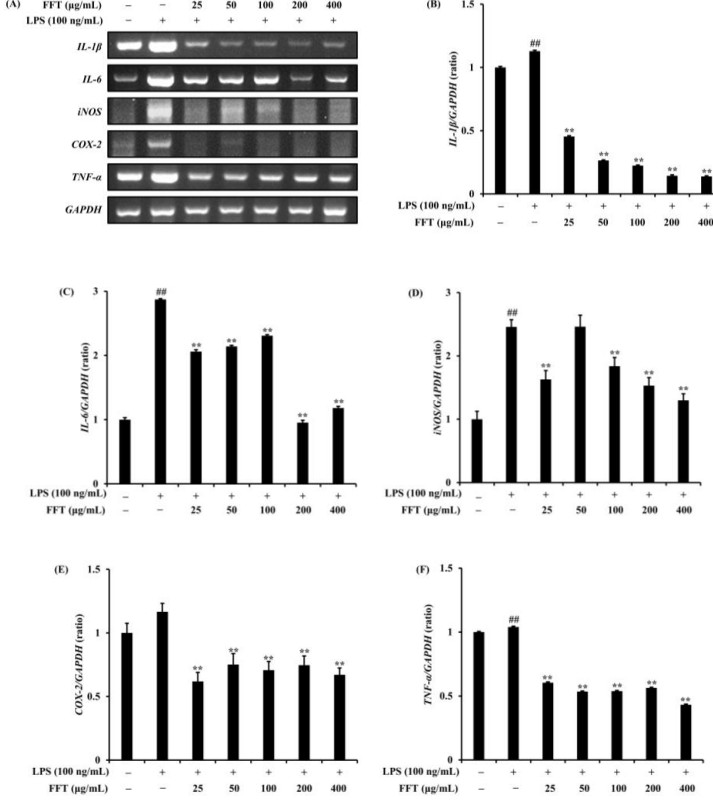

**Figure 5.** Effects of FFT on inflammatory cytokine mRNA expression in LPS-induced RAW 264.7 cells. RAW 264.7 cells were treated with various concentrations of FFT. (**A**) The results indicate inflammatory cytokine mRNA expression levels in LPS-induced RAW 264.7 cells. The graph indicates ratios of (**B**) IL-1β, (**C**) IL-6, (**D**) iNOS, (**E**) COX-2, and (**F**) TNF-α. GAPDH was used as a loading control. Data are expressed as means ± standard deviations (*n* = 3) ## *p* < 0.01 compared to the normal group. ** *p* < 0.01 compared to the LPS group. See also Figure S2.

### 3.5. Effects of FFT on NF-κB Phosphorylation and Nuclear Translocation

The influence of FFT on *nuclear factor kappa B* (NF-κB) nuclear translocation was investigated using Western blot analysis. FFT treatment led to a decrease in NF-κB nuclear translocation and phosphorylation in RAW 264.7 cells stimulated with LPS (Figure 6).

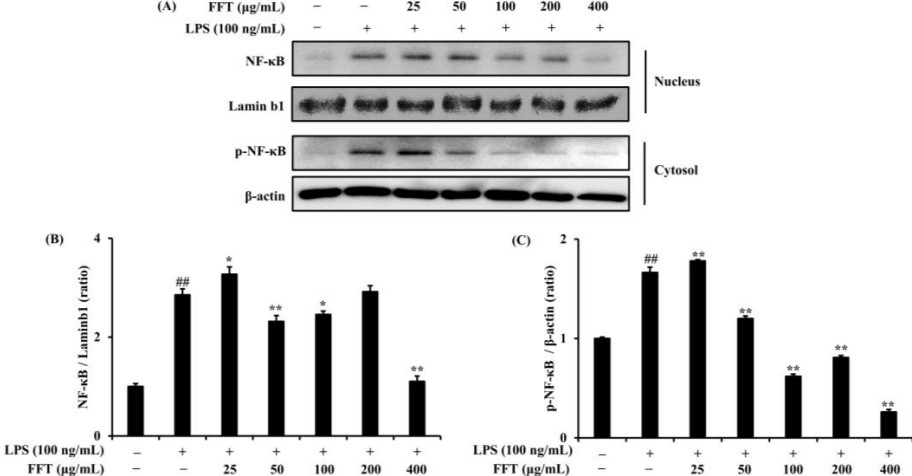

**Figure 6.** Effects of FFT on NF-κB and p-NF-κB in LPS-induced RAW 264.7 cells. RAW 264.7 cells were treated with various concentrations of FFT. (**A**) The results indicate NF-κB and p-NF-κB expression levels in the nucleus and cytoplasmic protein. The graph indicates ratios of (**B**) NF-κB and (**C**) p-NF-κB. Lamin b1 and GAPDH were used as loading controls. Data are expressed as means ± standard deviations (*n* = 3). ## *p* < 0.01 compared to the normal group. * *p* < 0.05 and ** *p* < 0.01 compared to the LPS group. See also Figure S3.

### 3.6. Effects of FFT on MAPK Signaling Pathway

The effects of FFT on the activated MAPK pathway were analyzed through Western blot assay. Results showed increased phosphorylation of ERK, JNK, and P38 in the LPS-treated group compared to the control group. FFT treatment significantly reduced JNK phosphorylation, and P38 phosphorylation decreased at 400 μg/mL. No significant effect was observed on ERK phosphorylation (Figure 7).

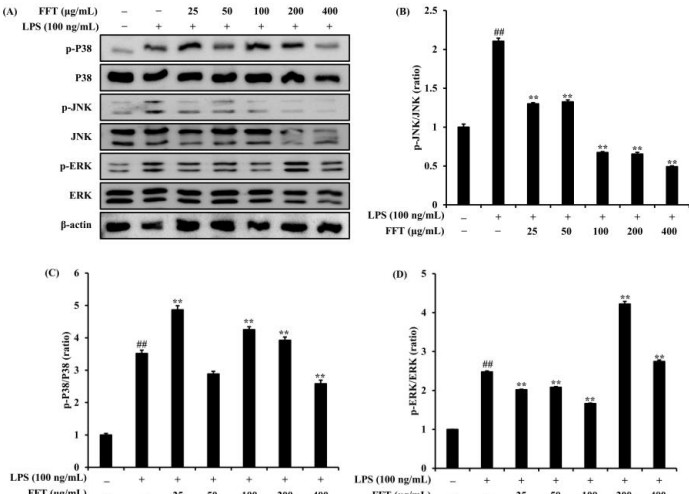

**Figure 7.** Effects of FFT on MAPK signal in LPS-induced RAW 264.7 cells. RAW 264.7 cells were treated with various concentrations of FFT and LPS (100 ng/mL) for 30 min. (**A**) The results represent MAPK signal protein expression levels. The graph indicates ratios of (**B**) p-P38, (**C**) p-JNK, and (**D**) p-ERK. Data are expressed as means ± standard deviations (*n* = 3). ## *p* < 0.01 compared to the normal group. ** *p* < 0.01 compared to the LPS group. See also Figure S4.

### 3.7. Effect of FFT on Lipid Accumulation in HepG2 Cells

To assess the cytotoxicity of FFT in HepG2 cells, cells were treated with FFA (1 mM) and different concentrations of FFT (50, 100, 200, and 400 μg/mL) for 48 h. Cell viability analysis revealed no significant toxicity at all concentrations (Figure 8A). Oil red O staining was performed to evaluate the inhibitory effect of FFT on lipid accumulation in HepG2 cells. Treatment with FFT at 400 μg/mL significantly reduced lipid accumulation compared to the FFA-treated group (Figure 8B,C). This reduction in lipid accumulation was consistent with the cell viability results, indicating an absence of cytotoxicity.

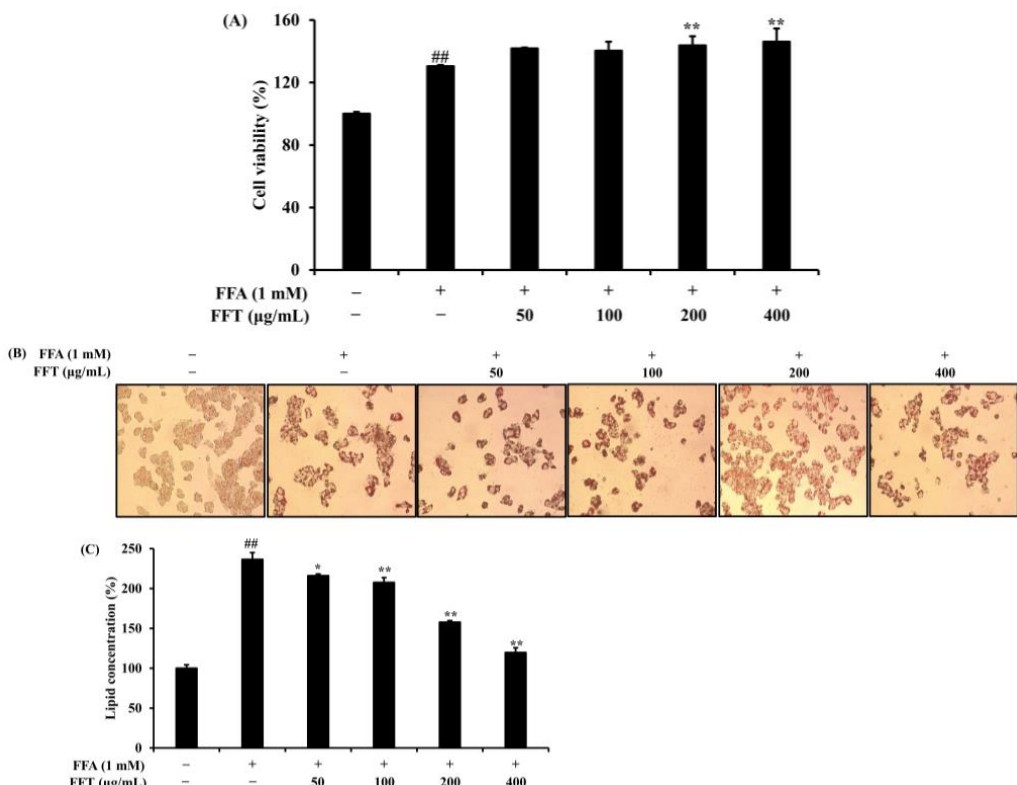

**Figure 8.** Effects of FFT on cell viability and lipid accumulation in FFA-induced HepG2 cells. HepG2 cells were treated with various concentrations of FFT and FFA (1 mM) for 48 h. (**A**) Cell viability was measured through MTT assay. The results indicate (**B**) microscope observation (200×) of cells and (**C**) intracellular lipid concentration after oil red O stain. Data are expressed as means ± standard deviations ($n = 3$). ## $p < 0.01$ compared to the normal group. * $p < 0.05$ and ** $p < 0.01$ compared to the FFA group.

### 3.8. Regulation of Lipogenesis-Related Protein Expression by FFT

Western blot analysis was conducted to analyze the expression of lipogenesis-related proteins in HepG2 cells treated with FFA and different concentrations of FFT. Results showed that compared to the normal group, FFA treatment increased the expression of CCAAT/enhancer binding protein (C/EBP)α and sterol regulatory element binding protein (SREBP)-1c while FFT treatment resulted in decreased expression (Figure 9).

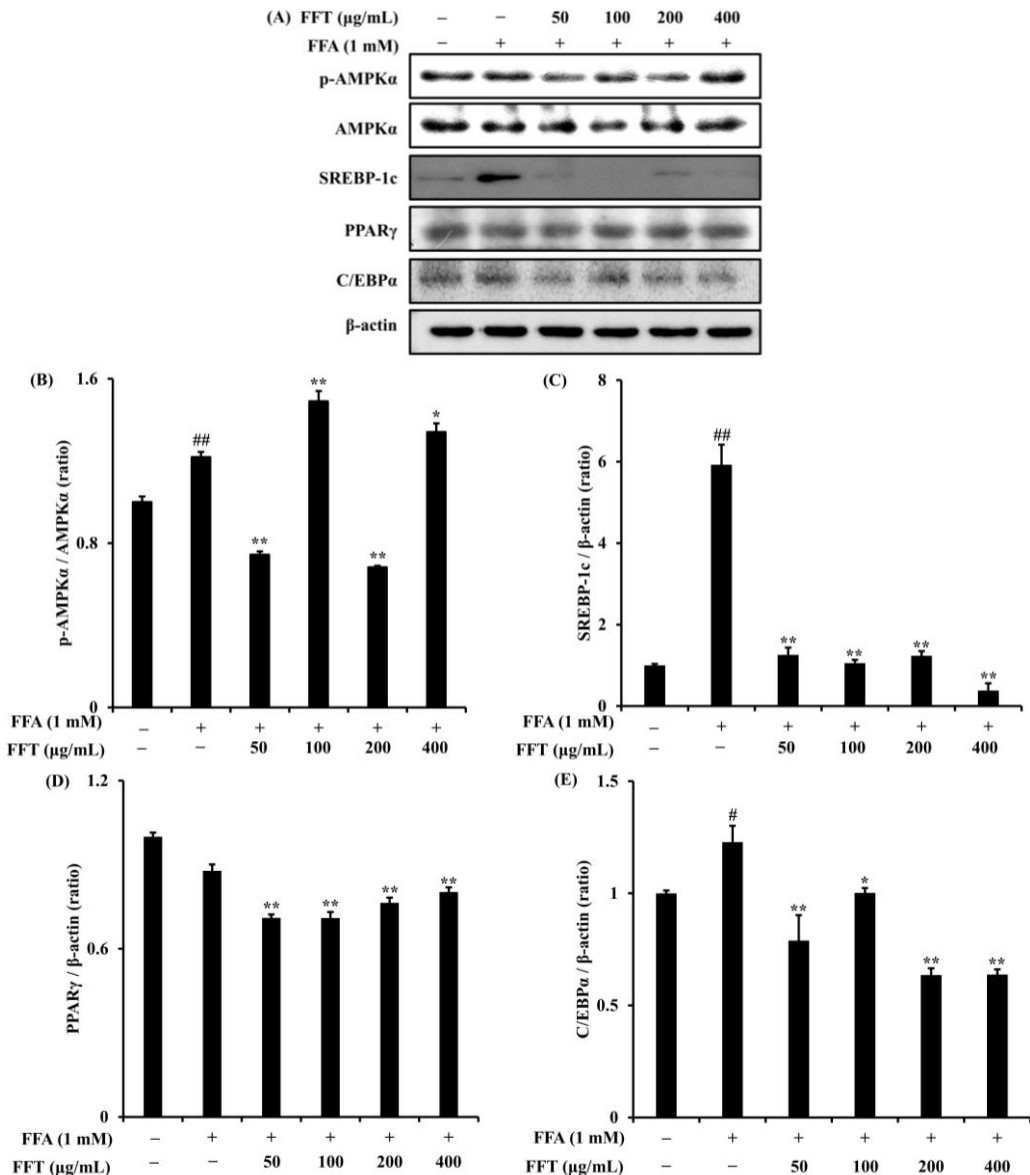

**Figure 9.** Effects of FFT on lipogenesis protein expression in FFA-induced HepG2 cells. HepG2 cells were treated with various concentrations of FFT and FFA (1 mM) for 48 h. (**A**) The results indicate lipogenesis protein expression levels. The graph indicates ratios of (**B**) p-AMPKα (**C**) SREBP-1c, (**D**) PPARγ, and (**E**) C/EBPα. β-actin and AMPKα were used as loading controls. Data are expressed as means ± standard deviations (*n* = 3). # *p* < 0.05 and ## *p* < 0.01 compared to the normal group. * *p* < 0.05 compared to the FFA group. ** *p* < 0.01 compared to the FFA group. See also Figure S5.

## 3.9. Effects of FFT on mRNA Expression of Lipogenesis-Related Genes

To evaluate the inhibitory effect of FFT on lipogenesis, the mRNA expression of lipogenesis-related genes was measured in HepG2 cells treated with FFA and FFT. Results showed that FFA treatment increased the expression of SREBP-1c and acetyl-CoA carboxylase (ACC)1 while FFT treatment reduced their expression (Figure 10). The observed reduction in lipogenesis-related gene expression suggests that FFT effectively inhibits lipogenesis in HepG2 cells.

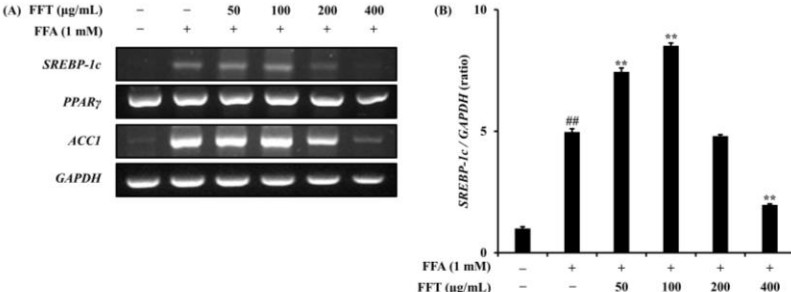

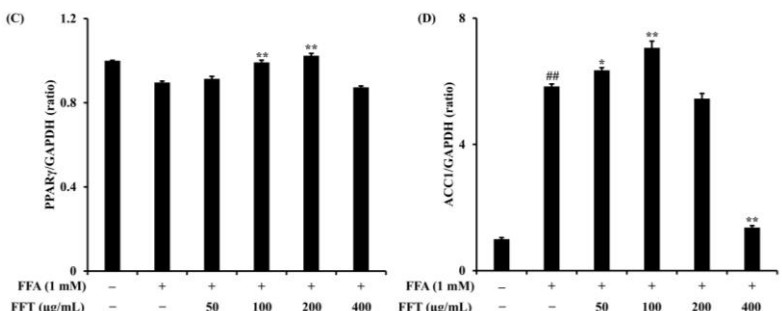

**Figure 10.** Effects of FFT on lipogenesis gene mRNA expression in FFA-induced HepG2 cells. HepG2 cells were treated with various concentrations of FFT and FFA (1 mM) for 48 h. The results represent (**A**) lipogenesis gene mRNA expression levels in FFA-induced HepG2 cells. The graph indicates ratios of (**B**) SREBP-1c, (**C**) PPARγ, and (**D**) ACC1. GAPDH was used as a loading control. Data are expressed as means ± standard deviations (*n* = 3). ## $p < 0.01$ compared to the normal group. * $p < 0.05$ compared to the FFA group. ** $p < 0.01$ compared to the FFA group. See also Figure S6.

## 4. Discussion

NAFLD is known to manifest with features such as cell death and inflammatory infiltration. Its primary causes are obesity and lipid synthesis due to the accumulation of neutral fats within the liver. Excessive fat accumulation in the liver can lead to conditions such as oxidative stress, inflammatory reactions, fibrosis, and cirrhosis. The early stage of NAFLD presents as simple steatosis, which can progress to steatohepatitis and, in the presence of inflammation, advance to cirrhosis [31,32].

Bioconversion is a technology that utilizes microbial or enzymatic biological reactions to generate various metabolites [33]. Fermentation in food is a prominent example of bioconversion, wherein beneficial microorganisms act on organic substances, transforming them into different forms [34]. Through fermentation, isoflavones, a type of flavonoid, can be converted to their aglycone forms, enhancing functionality [35]. It is known that during fermentation, *B. subtilis*, a common microorganism, produces various amylases and protein-degrading enzymes [36]. Protein degradation by these enzymes leads to the production of bioactive substances [37]. Therefore, *F. tataricum* Gaertner processed using bioconversion technology demonstrated superior inhibition of NO production compared to the negative control, along with an observed increase in the content of rutin, a representative component of *F. tataricum* Gaertner. In a study by Doe et al. [38], Raw 264.7 cells activated with 100 ng/mL LPS were treated with garlic non-fermented extract and fermented extract at various concentrations. The investigation revealed that fermented extract was more effective in inhibiting NO production compared to non-fermented extract, consistent with our findings. This confirms that fermentation enhances the inhibitory effect on NO production.

Rutin, a compound formed by combining rutinoside with quercetin, undergoes enzymatic degradation during the processing of buckwheat, resulting in decreased rutin levels and increased quercetin levels in sprouted *F. tataricum* Gaertner [39,40]. Consistent with prior research, this study suggests that the amount of rutin decreases during the

sterilization process and that the compounds generated through fermentation appear to transform quercetin into rutin.

Upon the activation of macrophages by LPS, iNOS generates NO through the metabolism of L-arginine. While NO plays a crucial role in host defense and maintaining cell function, its overproduction in response to external stimuli can promote inflammatory reactions [41,42]. Inflammatory responses involve the secretion of pro-inflammatory cytokines such as tumor necrosis factor (TNF)-$\alpha$, interleukin (IL)-6, and inflammatory mediators like TNF-$\alpha$, IL-6, and IL-1$\beta$ [43]. NF-$\kappa$B, a transcription factor complex containing p65 and p50, remains in an inactive form in the cytoplasm by binding with inhibitor kappa B (I$\kappa$B) under normal conditions. Upon inflammation, NF-$\kappa$B is activated, leading to its dissociation from I$\kappa$B and translocation to the nucleus, promoting the transcription of inflammatory cytokines such as TNF-$\alpha$, IL-6, IL-1$\beta$, iNOS, and COX-2 [8,44,45]. Macrophages stimulated by LPS activate the MAPK pathway through Toll-like receptor 4 (TLR4) [46]. The MAPK pathway includes independent routes like P38, JNK, and ERK, which, when phosphorylated due to LPS-induced inflammation, mediate inflammatory responses [47]. In this study, it was confirmed that FFT both regulates the expression of iNOS, COX-2, and inflammatory cytokines as well as inhibits the phosphorylation of NF-$\kappa$B and MAPK pathways, thus modulating inflammatory responses.

Oil red O staining, a common technique for staining neutral fats and lipoproteins, was utilized to observe a reduction in lipid accumulation upon treatment with FFT in HepG2 cells. Additionally, since FFT demonstrated no cytotoxicity at all tested concentrations, the observed reduction in lipid accumulation can be attributed to the sample's bioactivity. FFA entering the liver leads to the activation of SREBP-1c, inducing the activation of lipogenic enzymes such as ACC and fatty acid synthase (FAS). SREBPs act as crucial transcription factors regulating the expression of genes involved in fatty acid and neutral lipid synthesis. Moreover, the increased phosphorylation of AMP-activated protein kinase (AMPK) inhibits ACC and FAS, suppressing fatty acid synthesis [48–51]. Peroxisome proliferator-activated receptor (PPAR) is a nuclear hormone receptor involved in lipid and glucose metabolism [52]. PPAR$\gamma$, when binding to the gene regions associated with lipid synthesis, acts as a transcription factor, increasing the storage capacity of adipose tissue and decreasing the amount of free fatty acids in the body [53]. Given that the main cause of NAFLD is an increase in liver fat, the regulation of the expression of genes involved in liver fat synthesis is crucial for improving NAFLD. FFT was found to regulate lipid synthesis through the modulation of the expression of SREBP-1c, ACC, and C/EBP$\alpha$ in HepG2 cells induced by FFA.

This study suggests that FFT could be utilized as a therapeutic agent for NAFLD improvement by inhibiting lipid accumulation. However, considering the unclear onset mechanisms of NAFLD, further research is needed to confirm the activity of FFT against other variables that may contribute to its development.

## 5. Conclusions

This study confirmed the beneficial effects of fermented *F. tataricum* Gaertner extract on NAFLD. FFT demonstrated superior inhibition of NO production compared to NFT. In RAW 264.7 cells induced by LPS, FFT suppressed the expression of iNOS and COX-2 proteins, along with inhibiting the expression of inflammatory cytokine mRNA. The observed anti-inflammatory effects of FFT were attributed to its inhibition of phosphorylation in the NF-$\kappa$B and MAPK pathways. In HepG2 cells induced by FFA, FFT not only inhibited lipid accumulation but also regulated the expression of genes involved in lipid synthesis. These findings suggest the potential of FFT as a therapeutic agent for improving NAFLD, attributed to the increased presence of the physiologically active compound rutin. However, further research, particularly using animal models of NAFLD, is necessary to validate these effects and assess long-term safety and efficacy.

**Supplementary Materials:** The following supporting information can be downloaded at: https://www.mdpi.com/article/10.3390/fermentation10030116/s1, Figure S1: Original photographs for the blots of each protein marker of Figure 4A; Figure S2: Original photographs for the gels of each DNA marker of Figure 5A; Figure S3: Original photographs for the blots of each protein marker of Figure 6A; Figure S4: Original photographs for the blots of each protein marker of Figure 7A; Figure S5: Original photographs for the blots of each protein marker of Figure 9A; Figure S6: Original photographs for the gels of each DNA marker of Figure 10A.

**Author Contributions:** Conceptualization, S.-G.L. and H.K.; methodology, C.-H.P. and S.-G.L.; formal analysis, C.-H.P. and H.K.; investigation, S.-G.L. and C.-H.P.; resources, C.-H.P.; data curation, S.-G.L. and C.-H.P.; writing—original, C.-H.P.; writing—review and editing, S.-G.L. and H.K.; visualization, S.-G.L. and C.-H.P.; supervision, H.K.; project administration, S.-G.L.; funding acquisition, S.-G.L. All authors have read and agreed to the published version of the manuscript.

**Funding:** This work has supported by a National Research Foundation of Korea (NRF) grant, funded by the Korean government (MSIT) (No. 2021R1F1A1063617).

**Institutional Review Board Statement:** Not applicable.

**Informed Consent Statement:** Not applicable.

**Data Availability Statement:** Data are contained within the article and Supplementary Materials.

**Conflicts of Interest:** The authors declare no conflicts of interest.

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
