# Peer review of "Enhanced Anti-Inflammatory and Non-Alcoholic Fatty Liver Disease (NAFLD) Improvement Effects of Bacillus subtilis-Fermented Fagopyrum tataricum Gaertner"

_fermentation, doi:10.3390/fermentation10030116_

Round 1

Reviewer 1 Report

Comments and Suggestions for Authors

fermentation-2825876-peer-review-v1

The paper could be improved in some of its sections

a-    The title could be improved, include botanical Family and vulgar name of this specie

b-    The abstract should be improved in its writing; some paragraphs are difficult to understand.

It should also be more specific regarding the presentation of the results. A statistical support for the data should be mentioned.

c-    Introduction

The paragraph between lines 55-60 should be supported by additional bibliographic references.

d-    Introduction

Although various studies have been conducted on F. tataricum Gaertner ( page 2, line 66)

Mention such studies more specifically and include the respective bibliographic references.

Include a brief paragraph on their medicinal use (references)

Fagopyrum tataricum (L.) Gaertn.: A Review on its Traditional Uses, Phytochemical and Pharmacology

March 2017Food Science and Technology Research 23(1):1DOI: 10.3136/fstr.23.1 Lijuan LvYuan XiaDezhi Zou Minhui Li

e-    2.2. Preparation of NFT and FFT

Page 2; Lines 91-92

The F. tataricum Gaertner was ground, and 10 times the amount of distilled water (DW) was added for rehydration.

Mention which part of the plant has been used

How many grams of the species were used?

f-     2.3. High-Performance Liquid Chromatography (HPLC)

Include brand and company and country of origin of the HPLC equipment

g-    Table 2. Conditions for the HPLC analysis of the quercetin and rutin content in the NFT and FFT

Please review the table presentation

h-   3.1. Rutin and Quercetin Content in NFT and FFT

Please mention whether the quantification assays were done one, two, or three times or more. In which case present the results as the average value ± standard deviation.

It also mentions if there are significant differences between the samples, if I apply any statistics.

i-     Conclusion

Mention the role of quantified compounds, if relevant

The paper should be accepted after these suggestions made

Comments on the Quality of English Language

Moderate editing of English language required

Author Response

Response to Reviewer 1 Comments

Point 1: The title could be improved, include botanical Family and vulgar name of this specie.

Point 1 response:  We have added the botanical family and species name to the introduction. (Line 48-49)

Point 2:
The abstract should be improved in its writing; some paragraphs are difficult to understand. It should also be more specific regarding the presentation of the results. A statistical support for the data should be mentioned.

Point 2 response: In response to the reviewer's feedback, we have refined the abstract to enhance clarity, specificity in presenting results, and included statistical support for the data. (Line 10-29)

Point 3: Introduction

3-1: The paragraph between lines 55-60 should be supported by additional bibliographic references.

Point 3-1 response: In response to the reviewer's comment, the paragraph between lines 55-60 has been strengthened with additional bibliographic references for a more robust foundation. (Line 54-65, 424-442)

3-2: Although various studies have been conducted on F. tataricum Gaertner (page 2, line 66)

Mention such studies more specifically and include the respective bibliographic references.

Include a brief paragraph on their medicinal use (references) Fagopyrum tataricum (L.) Gaertn.: A Review on its Traditional Uses, Phytochemical and Pharmacology March 2017Food Science and Technology Research 23(1):1DOI: 10.3136/fstr.23.1 Lijuan LvYuan XiaDezhi Zou Minhui Li

Point 3-1 response: Incorporating the reviewer's feedback, I have inserted additional references providing insights into the various medicinal uses of F. tataricum Gaertner. (Line 69, 417,418)

Point 4: 2.2. Preparation of NFT and FFT

4-1: Page 2; Lines 91-92

The F. tataricum Gaertner was ground, and 10 times the amount of distilled water (DW) was added for rehydration.

Mention which part of the plant has been used

How many grams of the species were used?

Point 4-1 response: The F. tataricum Gaertner grain was ground, and 100 grams of it were used for the experiment. This detailed information has been added to provide clarity on the plant part utilized and the quantity used in the study. (Line 94-104)

4-2: High-Performance Liquid Chromatography (HPLC)

Include brand and company and country of origin of the HPLC equipment

Point 4-2 response: As per the reviewer's suggestion, the information has been added. (Line 109-110)

Point 5: Table 2. Conditions for the HPLC analysis of the quercetin and rutin content in the NFT and FFT

Please review the table presentation

Point 5 response:  We have revised as a reviewer's suggestions. (Line 117)

Point 6: 3.1. Rutin and Quercetin Content in NFT and FFT

Please mention whether the quantification assays were done one, two, or three times or more. In which case present the results as the average value ± standard deviation.

It also mentions if there are significant differences between the samples, if I apply any statistics.

Point 6 response:  As per the reviewer's suggestion, the information has been added. (Line 175-183)

Point 7: Conclusion

Mention the role of quantified compounds, if relevant

Point 7 response:  As per the reviewer's suggestion, the information has been added. (Line 372-375)

Reviewer 2 Report

Comments and Suggestions for Authors

In the manuscript “Enhanced Anti-Inflammatory and Non-Alcoholic Fatty Liver Disease (NAFLD) Improvement Effects of Bacillus subtilis Fermented Fagopyrum tataricum Gaertner” authors study the biological effect of an alcoholic extract obtained after fermentation.

 The topic is very interesting and fits within the scope of the journal. Nevertheless, the article has many flasks that must be addressed before publication.

 Major flasks are in the methodology description. Below authors can find some commentaries.

 Abstract

Line 12: authors say: “The non-fermented F. tataricum Gaertner extract (NFT) and the active components, rutin and quercetin in FFT”. It seems like NFT do not have rutin or quercetin, which is not true, since both compounds are present in the whole plant.

Introduction

Line 37: which is the meaning of “domestically”?

Lines 41-43 repeat what was said in lines 33-36.

 Line 47: which is the meaning of: “byproducts of respiratory processes in the body”.

Line 54: which type of “carbohydrates”?

Line 58: which type of components are “rutin, quercetin, quercetrin, and catechins”?

Line 65: which type of “functional properties” have rutin?

Materials and methods

Line 91: "ground"? Or grown? Or harvested?

Line 92: how was sterilized the mixture???? Heated? The compounds of interests do not degrade in the autoclave???  

Line 93: in which proportion was added Bacillus subtilis? In which volume???

and: how was the microorganism cultured?  in which media?

Line 93: the extracts was obtained without removing que bacteria?

Line 97: The affirmation was prepared by inoculating the substrate is not clear for this referee.

Line 104: “Samples were dissolved in DMSO” pure?

Lines 102-109: did authors used any reference substance (pattern) to assign peaks to rutin and quercetin?

Line 121: authors did read the absorbance of MTT directly? Or they incubated with isopropanol or DMSO to read the absorbance? In which equipment did they read the absorbance? In a spectrophotometer? In a plate reader?

References of methodologies are needed (i.e.: MTT assay).

Line 124: which is the purpose of the addition od oleic and palmitic acid in the culture medium?

Methodologies described in items 2.7 and 2.8 need references.

Comments on the Quality of English Language

language must be improved

Author Response

Response to Reviewer 2 Comments

Point 1: Line 12: authors say: “The non-fermented F. tataricum Gaertner extract (NFT) and the active components, rutin and quercetin in FFT”. It seems like NFT do not have rutin or quercetin, which is not true, since both compounds are present in the whole plant.

Point 1 response:  We have revised it as 'marker components.' (Line 13-14)

Point 2: Line 37: which is the meaning of “domestically”?

Point 2 response: We have made revisions. (Line 38-39)

Point 3: Lines 41-43 repeat what was said in lines 33-36.

Point 3 response: We have removed the duplicated content. (Line 34-40)

Point 4: Line 47: which is the meaning of: “byproducts of respiratory processes in the body”.

Point 4 response: We have made revisions. (Line 43-44)

Point 5: Line 54: which type of “carbohydrates”?

Point 5 response: We have made revisions. (Line 50-51)

Point 6: Line 58: which type of components are “rutin, quercetin, quercetrin, and catechins”?

Please review the table presentation

Point 6 response:  We have made revisions. (Line 55)

Point 7: Line 65: which type of “functional properties” have rutin?

Point 7 response:  The types of 'functional properties' associated with rutin have been explained in lines 59-65.

Point 8: Line 91: "ground"? Or grown? Or harvested?

Point 8 response:  "Grind" mean

Point 9: Line 92: how was sterilized the mixture???? Heated? The compounds of interests do not degrade in the autoclave??? 

Point 9 response:  The mixture was sterilized using an autoclave. While sterilization during fermentation can potentially degrade compounds, it is necessary to remove contaminants other than fermentation strains.

Point 10: Line 93: in which proportion was added Bacillus subtilis? In which volume??? and: how was the microorganism cultured?  in which media?

Point 10 response: Bacillus subtilis (KCTC 3014) was cultured in a broth medium (Kisanbio Co., Seoul. Korea) with a pH of 7.2. These microorganisms were cultured at 30℃ and used for the experiment. They were inoculated at a 10% concentration. (Line 94-98)

Point 11: Line 93: the extracts was obtained without removing que bacteria?

Point 11 response: The extracts were obtained after removing the bacteria through the filtration process.

Point 12: Line 97: The affirmation was prepared by inoculating the substrate is not clear for this referee.

Point 12 response: When we prepared NFT, we inoculated nutrient broth instead of B. subtilis.  Substrate means nutrient broth.

Point 13: Line 104: “Samples were dissolved in DMSO” pure?

Point 13 response: Samples were dissolved in HPLC grade DMSO.

Point 14: Lines 102-109: did authors used any reference substance (pattern) to assign peaks to rutin and quercetin?

Point 14 response We used quercetin (Q0125, Sigma) and rutin (R5143, Sigma) as reference substances. (Line 108,109)

Point 15: Line 121: authors did read the absorbance of MTT directly? Or they incubated with isopropanol or DMSO to read the absorbance? In which equipment did they read the absorbance? In a spectrophotometer? In a plate reader?

Point 15 response: We have revised the method for measuring cell viability in detail. (Line 125-135)

Point 16: References of methodologies are needed (i.e.: MTT assay).

Point 16 response: The references for methodologies have been incorporated, including the MTT assay. (Line 125)

Point 17: Line 124: which is the purpose of the addition od oleic and palmitic acid in the culture medium?

Point 17 response: To make a fatty acid-induced NAFLD model in HepG2 cell, we used oleic and palmitic acid

Point 18: Methodologies described in items 2.7 and 2.8 need references.

Point 18 response: Additional references for methodologies have been included.  (Line 143, 151)

Reviewer 3 Report

Comments and Suggestions for Authors

Thank you very much for your interesting research. Some points must be carefully revised:

INTRODUCTION. As well as some statements are included to discuss the biological activities of buckwheat phenolics, a paragraph must be added briefly describing the importance of bioactive carbohydrates/polysaccharides (references: 10.1016/j.ijbiomac.2019.01.043; 10.1016/j.foodchem.2020.127653) and peptides (reference: 10.1080/10408398.2020.1761774).

MATERIALS & METHODS. Materials and preparation of NFT and FFT: was F. tataricum material acquired as dry mass? Was it dried at the laboratory? Using temperature/lyophilization? Or was it preserved with a low moisture content and then rehydrated? Please, include these data.

RESULTS. Figure 3. Comparisons between NFT and FFT would be also interesting (statistical analysis).

DISCUSSION. Further comparison with previously published results is required in this section.

CONCLUSIONS. Please, included current limitations and future perspectives.

Author Response

Response to Reviewer 3 Comments

Point 1: INTRODUCTION. As well as some statements are included to discuss the biological activities of buckwheat phenolics, a paragraph must be added briefly describing the importance of bioactive carbohydrates/polysaccharides (references: 10.1016/j.ijbiomac.2019.01.043; 10.1016/j.foodchem.2020.127653) and peptides (reference: 10.1080/10408398.2020.1761774).

Point 1 response:  As per the reviewer's feedback, the introduction has been enhanced to include discussions on the biological activities of buckwheat phenolics, as well as the importance of bioactive carbohydrates/polysaccharides and peptides. (Line 59-65)

Point 2: MATERIALS & METHODS. Materials and preparation of NFT and FFT: was F. tataricum material acquired as dry mass? Was it dried at the laboratory? Using temperature/lyophilization? Or was it preserved with a low moisture content and then rehydrated? Please, include these data.

Point 2 response: As per the reviewer's feedback, the Materials and preparation of NFT and FFT have been revised. (Line 94-104)

Point 3: RESULTS. Figure 3. Comparisons between NFT and FFT would be also interesting (statistical analysis).

Point 3 response: As per the reviewer's feedback, the Figure 3 has been revised accordingly. (Line 193)

Point 4: DISCUSSION. Further comparison with previously published results is required in this section.

Point 4 response: As per the reviewer's suggestion, we have further compared our results with previously published findings.

Point 5: CONCLUSIONS. Please, included current limitations and future perspectives.

Point 5 response: As per the reviewer's suggestion, we have been revised. (Line 372-375)

Round 2

Reviewer 2 Report

Comments and Suggestions for Authors

Dear authors: Thank you for conducting and answering most of the questions made by this referee. The manuscript is able for publication now, in my opinion. 

Nevertheless, answers to points 9, 11 and 17 should be added to the manuscript.

I copy them below, in order to facilitate the revision:

Point 9: Line 92: how was sterilized the mixture???? Heated? The compounds of interests do not degrade in the autoclave??? 

Point 9 response:  The mixture was sterilized using an autoclave. While sterilization during fermentation can potentially degrade compounds, it is necessary to remove contaminants other than fermentation strains.

Point 11: Line 93: the extracts was obtained without removing que bacteria?

Point 11 response: The extracts were obtained after removing the bacteria through the filtration process.

Point 17: Line 124: which is the purpose of the addition od oleic and palmitic acid in the culture medium?

Point 17 response: To make a fatty acid-induced NAFLD model in HepG2 cell, we used oleic and palmitic acid

Moreover, this methodology of adding fatty acids to induce NAFLD needs a reference.

If the three points are conducted, the manuscript will be able for publication, in my opinion.

Author Response

Reviewer: Answers to points 9, 11 and 17 should be added to the manuscript.

Response: We have made the requested revisions.

Point 9: Line 92: how was sterilized the mixture???? Heated? The compounds of interests do not degrade in the autoclave??? 

Point 9 response:  The mixture was sterilized using an autoclave. While sterilization during fermentation can potentially degrade compounds, it is necessary to remove contaminants other than fermentation strains.

  • Line 97-99

Point 11: Line 93: the extracts was obtained without removing que bacteria?

Point 11 response: The extracts were obtained after removing the bacteria through the filtration process.

  • Line 103-104

Point 17: Line 124: which is the purpose of the addition od oleic and palmitic acid in the culture medium?

Point 17 response: To make a fatty acid-induced NAFLD model in HepG2 cell, we used oleic and palmitic acid

  • Line 137-139

Moreover, this methodology of adding fatty acids to induce NAFLD needs a reference.

  • Line 140, 451-452
